# A Performance-Oriented Optimization Framework Combining Meta-Heuristics and Entropy-Weighted TOPSIS for Multi-Objective Sustainable Supply Chain Network Design

**Yurong Guo, Quan Shi \* and Chiming Guo**

Shijiazhuang Campus, Army Engineering University, Shijiazhuang 050003, China
* Correspondence: gyr2546472152@126.com

**Abstract:** The decision-making of sustainable supply chain network (SSCN) design is a strategy capacity for configuring network facility and product flow. When optimizing conflicting economic, environmental, and social performance objectives, it is difficult to select the optimal scheme from obtained feasible decision schemes. In this article, according to the triple bottom line of sustainability, a multi-objective sustainable supply chain network optimization model is developed, and a novel performance-oriented optimization framework is proposed. This framework, referred to as performance-oriented optimization framework, integrates multi-objective meta-heuristic algorithms and entropy-weighted technique for order preference by similarity to an ideal solution (EW-TOPSIS). The optimization framework can comprehensively evaluate the performance of overall SSCN by EW-TOPSIS and guide the evolution process of algorithms. In this framework, decision-makers can obtain the feasible schemes calculated by meta-heuristics and determine the optimal one according to the performance value evaluated by EW-TOPSIS. This article combines three performance evaluation strategies with four meta-heuristic algorithms, namely, non-dominated Sorting Genetic Algorithm-II (NSGA-2), multi-objective differential evolutionary (MODE), multi-objective particle swarm optimization (MOPSO), and multi-objective gray wolr optimization (MOGWO), for verifying the effectiveness of the performance-oriented optimization framework. The results validate that the proposed framework has much better sustainability performance than the traditional optimization algorithms and evaluation methods. Furthermore, the proposed performance-oriented optimization framework can provide managers with a special optimal scheme with the best sustainability performance. Finally, some research prospects are presented such as more multi-criteria decision making methods.

**Keywords:** sustainable supply chain; performance evaluation; entropy-weighted TOPSIS; meta-heuristic; multi-objective optimization

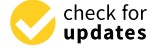



## 1. Introduction

Recently, sustainable supply chain network design (SSCND), which consists of economic development, environmental protection and social responsibility, has played an important role in improving the performance and efficiency of a supply chain,. SSCND spans strategy decisions, tactical decisions and operational decisions involving facilities location, quantity and capacity of facilities, selection of transportation modes and product flow between facilities. However, the strategy decision, which indicates the configuration of the supply chain network, cannot be changed on the whole horizon, because the economic and time cost of network change are huge. On the other hand, the tactical decisions, such as product flow and facilities capacity, can be adjusted with the change in production periods or various scenarios.

For many years, supply chain network design (SCND) has primarily aimed to minimize the total cost or maximize the total profit of a supply chain for seeking maximized economic performance, without taking into consideration environmental pollution or the

health of employees. However, recently, with the increasing awareness of the need to protect the ecological environment and safeguard the rights of employees, non-governmental organizations and social media are demanding that enterprises should take responsibility for their productions and operations in their supply chain network [1]. The focus on ecological and social impacts of supply chain operation has led to the development of a novel paradigm named "sustainability". It is becoming common and critical for supply chain managers to take into consideration sustainable performance in SCND. The concept of "sustainability" reminds us that present development should consider and not compromise the needs of future generations. Although simple in the above definition, the way to realize sustainable development is complex and difficult in real supply chain activities. For operationalizing sustainability in supply chain activities, a central concept called the triple bottom line is proposed, which divides sustainability into three dimensions: the business case (economic), the societal case (social) and the natural case (environmental) [2–5]. The triple bottom line of economic, environmental and social performance are interconnected. Most of the research has proposed that there is a positive correlation between reducing environmental pollution, assuming social responsibility and improving economic benefit [6,7]. Compared to the traditional supply chain network, the sustainable supply chain can not only obtain the dual economic and environmental performance, but also seek an excellent trade-off between economic, social and ecological performance [8].

In order to respond to the sustainability paradigm, some studies have begun to take into consideration sustainable supply chain management (SSCM) in SSCND. However, the research on SSCND that quantitatively models all three dimensions of sustainability is limited [9]. Moreover, most research in the field of SSCM only pays attention to the responsive and resilient feature of the SSCN but fails to take into consideration a critical characteristic of sustainable performance measurement. Sustainable supply chain performance measurement can be defined as the process of measuring the efficiency and influence of an action or an object quantitatively and/or qualitatively [10,11]. The main purpose of SSCPM is to determine the best among existing alternative schemes [12]. It is critical to determine appropriate indicators for evaluating sustainability performance. Measurement indicators are utilized to represent the state of economical, natural, and social development in the supply chain network system [13]. Nevertheless, because too many indicators will lead to inconsistency of dimensions, it is difficult to measure supply chain sustainability performance. Therefore, the problem is that all valuations should be simplified into a single one-dimensional standard in certain ways. To address the above problem, multi-criteria decision-making (MCDM) introduces a solution framework. In MCDM, all measurement indicators are presented in their original form. That is to say, the MCDM approach provides the most constructive solution framework for assessing sustainability performance.

It can be concluded that sustainable supply chain optimization can obtain some feasible schemes for optimizing strategy and tactical decisions, and MCDM can evaluate supply chain sustainability performance for selecting the optimal decision scheme. Nevertheless, most academic studies divide supply chain network optimization and sustainability performance measurement [14]. In the author's view, optimization and evaluation are inseparable in SSCND. Mandal et al. also support this view, and they adopted the TOPSIS method to evaluate the obtained results calculated by multi-objective meta-heuristic algorithms in multi-objective machining problems [15]. In an attempt to integrate network optimization and performance measurement, a performance-oriented optimization framework is developed. According to this framework, a set of feasible solutions can be obtained by optimization approaches and sorted by MCDM methods. The optimal solution can be selected based on evaluation efficiency values calculated by special evaluation methods. The greatest difference with Mandal's thought is that the optimization process and performance measurement are performed simultaneously, and the network optimization is guided by sustainability measurement.

The structure of this article is as follows: Section 2 reviews some literature involving supply chain network optimization, multi-criteria decision making and the integration of

network optimization and performance evaluation. Section 3 formulates a multi-objective mixed-integer linear programming model. In Section 4, a performance-oriented optimization framework is developed. Section 5 represents the result and discussion related to a numerical example. In Section 6, the conclusions are elaborated.

## 2. Literature Review

### 2.1. Supply Chain Network Optimization

Supply chain network optimization can be classified as a mathematical programming problem. Multi-objective optimization based on triple bottom line indicators is widely used to model the SSCND problem. Nayeri et al. consider economic, environmental, and social aspects as optimization objectives for designing SCLSC network [16]. A multi-objective mixed-integer programming (MOMIP) model was developed, and multi-choice goal programming with the utility function was used to solve the MOMIP model. To broaden the dimensions of supply chain features, they introduced responsive indicators and resilient indicators into the supply chain network for optimizing a sustainable, responsive, and resilient supply chain [17]. Zhang et al. formulated a MOMIP model considering economic performance and social aspects [18]. The lost working days caused by work damages modeled by fuzzy programming are considered social indicators. Pourmehdi et al. formulated a multi-objective stochastic programming model to design a sustainable steel supply chain [19]. They considered the total cost as economic performance and emission in production as the environmental aspect. Soleimani et al. modeled a green sustainable closed-loop supply chain by considering environmental protection, profit optimization, and a decrease in lost workdays caused by occupational accidents [20]. The improved genetic algorithm was adopted to solve their developed model. Abad et al. proposed a novel bi-objective chance-constraint programming approach for dealing with the uncertain green supply chain [21]. The first objective aims to minimize the total cost, and the second objective controls the scatter of uncertain cost, which is described as the variance in stochastic variable cost. Fazli-Khalaf et al. formulated a bi-objective programming model for a green reliable supply chain, which aims to minimize total operating cost and maximize the greenness of the designed supply chain network [22]. They developed a novel fuzzy robust stochastic optimization technical for coping with hybrid uncertainty in the supply chain network.

### 2.2. Supply Chain Performance Evaluation

MCDM methods are widely used in supply chain management, mainly consisting of supplier selection and sustainability measurement. Supplier selection is an important strategic decision for sustainable supply chain management, which is regarded as a process of MCDM [23,24]. Sustainability measurement is usually performed to evaluate sustainability performance in terms of multiple aspects within the triple bottom line framework [12]. The purpose of adopting MCDM methods is to find the optimal scheme in the existing feasible schemes for performing supplier selection and sustainability performance evaluation [23,25,26]. Erol et al. divided supply chain sustainability performance evaluation into economic, environmental and social aspects on the basis of two criteria: "an indicator's measurability" and "data availability for a special indicator" [27]. Ahi et al. reviewed the measure indicators in green supply chain and sustainable supply chain management [11]. The most often used metrics are quality, greenhouse gas emission, air emission, energy use and energy consumption. They proposed that the economic, environmental and social aspects are the most efficient indicators used to measure supply chain sustainability performance. Tavassoli et al. developed a double-frontier fuzzy network data envelopment analysis evaluation model for measuring the sustainability performance of the tomato paste supply chain [28]. Wang et al. proposed an efficiency sorting multi-objective optimization framework to make optimization solutions and decision-making on a sustainable supply chain [22]. In this framework, data envelopment analysis (DEA) was utilized to measure supply chain performance, and multi-objective optimization algorithms optimized network

design. However, DEA is an MCDM based on input indicators and output indicators. If all the selected indicators are minimized or maximized, it is difficult to evaluate supply chain performance by DEA.

TOPSIS, as a commonly used MCDM method, is a technique used to sort finite evaluation objects according to their proximity to the ideal solution, which has been broadly adopted to evaluate supply chain sustainability performance and suppliers. Sun et al. introduced entropy-weighted TOPSIS to assign objective weight values to each evaluation indicator [29]. Li et al. developed a rough cloud TOPSIS approach for supplier selection in a sustainable supply chain [23]. Marzouk et al. adopted AHP to calculate the relative importance weight of suppliers and TOPSIS to measure different suppliers in a constructive supply chain based on 17 indicators [26]. Venkatesh et al. developed a fuzzy AHP-TOPSIS method to deal with the complexity in the continuous aid supply chain [30]. Jellali et al. adopted a fuzzy method for TOPSIS to evaluate the sustainability of the olive oil supply chain [31]. Moreover, based on a review of MCDM, the most popular MCDM methods are AHP and TOPSIS [32]. However, the AHP method is based on expert judgment, so it is subjective, but the TOPSIS model is a completely quantitative and objective MCDM method for evaluating sustainability performance. Hence, in this article, the entropy-weighted TOPSIS method is utilized to evaluate the sustainability performance of decision schemes. Finally, it is critical to determine appropriate evaluation indicators to measure supply chain performance [12].

### 2.3. Integrating of Network Optimization and Performance Evaluation

By solving multi-objective supply chain network optimization problems on the basis of the triple bottom line, the feasible Pareto optimal solution set will be obtained, and it is difficult to select the optimal scheme from these alternative schemes. The process of achieving the optimal scheme can be regarded as network decision-making [12]. Little research has integrated supply chain network optimization and network decision-making. The common thought in the existing literature is to evaluate and sort the alternative schemes so as to obtain the optimal scheme with the best evaluation efficiency value.

Validi et al. used three independent optimization algorithms based on genetic algorithms (NSGA-2, MOGA-2 and HYBRID) to solve their green multi-objective model [33]. They paid attention to economic performance by minimizing cost and ecological performance by minimizing carbon emissions. Then, using a multi-attribute decision-making approach, TOPSIS, the solution is adopted to sort obtained alternative transportation routes. Mandal et al. adopted TOPSIS to evaluate feasible Pareto solutions calculated by MOPSO and select the optimal one with the best efficiency value in optimizing Cu-MWCNT composite electrode machining [15]. The solution decision-making approach in the above-mentioned literature can be summarized as a two-stage technical approach. In the first stage, heuristic methods are used to obtain the feasible decision scheme set (i.e., the final Pareto optimal solution set). In the second stage, various multi-criteria decision-making approaches are adopted to evaluate and sort these Pareto optimal schemes so as to obtain the optimal scheme. Furthermore, some studies focus on performing supplier measurement in the first stage and solution decisions in the second stage. Lahri et al. integrated supplier selection and SSCN design [34]. BMW and TOPSIS are utilized to measure the sustainability performance of suppliers so as to obtain the green image weight, which is used to design network design. The ε-constraint method is adopted to solve their multi-objective possibility mixed-integer programming model. Moheb-Alizadeh et al. utilized the bi-objective DEA method to measure the efficiency of supply chain nodes and considered them as the optimization objectives [35]. Then, the proposed multi-objective stochastic programming model was solved by the Lagrangian relaxation algorithm and multi-choice goal programming. It can be seen that two-stage methods in most studies are independent of each other; in other words, model solution and measurement are independent. Hence, a parallel approach, integrating optimization and measurement, is proposed in this research. In each iteration, the feasible solutions in each archive are sorted according to their sustainability

performance, and then, the sorted individuals guided the next iteration. In this article, the entropy-weighted TOPSIS method and meta-heuristic algorithms are integrated with the proposed performance-oriented optimization framework.

## 3. Model Formulation

### 3.1. Problem Description

This article examines a generic multi-echelon SSCN, which handles the forward flow of new products and the reverse flow of end-of-life (EOL) products. The structure of the generic SSCN structure is illustrated in Figure 1. The proposed network structure can be modeled in various industries, including the pharmaceutical industry [36], the steel-based products industry [37] and the tanker industry [17]. In Figure 1, the new products are produced by manufacturing centers and transited to distribution centers and then to the customer markets to satisfy their demands. Furthermore, when the products reach the end of life, the EOL products are collected in recycling centers where they are tested, inspected, sorted and classified into two different parts: unrecyclable EOL products and recyclable EOL products. The recyclable EOL products are reprocessed and assembled in recycling centers and returned to the customer market. On the other hand, unrecyclable EOL products are transited to disposal centers to be disposed of and scrapped by decomposition and/or incineration.

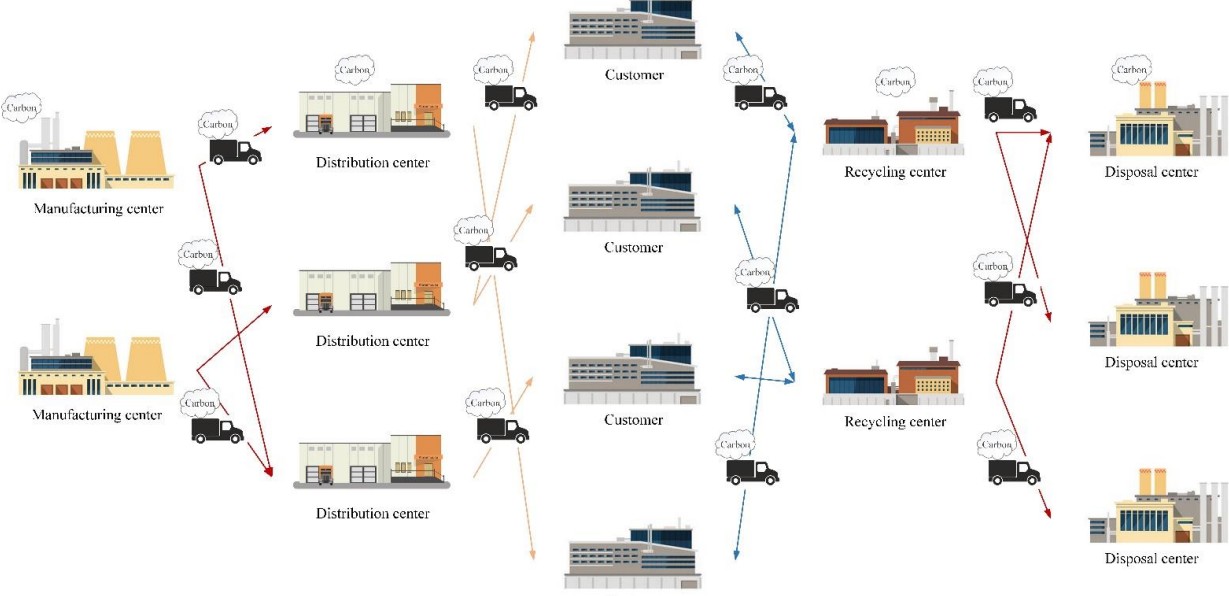

**Figure 1.** The structure of the generic sustainable supply chain network.

To specify facility location and product flow in the aforementioned SSCN, a MOMIP model is developed in this article. The proposed multi-objective model aims to minimize the total cost (economic performance), minimize the pollution emission (environmental performance) and maximize social responsibility by maximizing generated job opportunities and minimizing lost working days.

### 3.2. Model Symbols

The corresponding indexes, notations and parameters of the MOMIP model are shown as follows.

Indexes

| | |
|---|---|
| $i$ | Index of manufacturing centers, $i = 1, 2, \ldots, I$ |
| $j$ | Index of distribution centers, $j = 1, 2, \ldots, J$ |
| $k$ | Index of customers, $k = 1, 2, \ldots, K$ |
| $m$ | Index of recycling centers, $m = 1, 2, \ldots, M$ |
| $n$ | Index of disposal centers, $n = 1, 2, \ldots, N$ |
| $v$ | Index of disposal centers, $n = 1, 2, \ldots, N$ |

Parameters

| | |
|---|---|
| $PR_i$ | Price of product from manufacturing $i$ |
| $PR_m$ | Price of product from recycling center $m$ |
| $CT_{ijv}$ | Transportation cost of product per unit weight from manufacturing center $i$ to distribution center $j$ by transportation mode $v$ |
| $CT_{jkv}$ | Transportation cost of product per unit weight from distribution center $j$ to customer $k$ by transportation mode $v$ |
| $CT_{kmv}$ | Transportation cost of product per unit weight from customer $k$ to recycling center $m$ by transportation mode $v$ |
| $CT_{mkv}$ | Transportation cost of product per unit weight from recycling center $m$ to customer $k$ by transportation mode $v$ |
| $CT_{mnv}$ | Transportation cost of product per unit weight from recycling center $m$ to disposal center $n$ by transportation mode $v$ |
| $CH_i$ | Inventory holding cost of unit product in manufacturing center $i$ |
| $CH_j$ | Inventory holding cost of unit product in distribution center $j$ |
| $CH_m$ | Inventory holding cost of unit product in recycling center $m$ |
| $CH_n$ | Inventory holding cost of unit product in disposal center $n$ |
| $CP_i$ | Processing cost of unit product in manufacturing center $i$ |
| $CP_j$ | Processing cost of unit product in distribution center $j$ |
| $CP_m$ | Processing cost of unit product in recycling center $m$ |
| $CP_n$ | Processing cost of unit product in disposal center $n$ |
| $CR_k$ | Recycling price of the product in customer $k$ |
| $ET_{ijv}$ | Unit transportation pollution emission from manufacturing center $i$ to distribution center $j$ by transportation mode $v$ |
| $ET_{jkv}$ | Unit transportation pollution emission from distribution center $j$ to customer $k$ by transportation mode $v$ |
| $ET_{kmv}$ | Unit transportation pollution emission from customer $k$ to recycling center $m$ by transportation mode $v$ |
| $ET_{mkv}$ | Unit transportation pollution emission from recycling center $m$ to customer $k$ by transportation mode $v$ |
| $ET_{mnv}$ | Unit transportation pollution emission from recycling center $m$ to disposal center $n$ by transportation mode $v$ |
| $EP_i$ | Unit processing pollution emission in manufacturing center $i$ |
| $EP_j$ | Unit processing pollution emission in distribution center $j$ |
| $EP_m$ | Unit processing pollution emission in recycling center $m$ |
| $EP_n$ | Unit processing pollution emission in disposal center $n$ |
| $EH_i$ | Unit holding pollution emission in manufacturing center $i$ |
| $EH_j$ | Unit holding pollution emission in distribution center $j$ |
| $EH_m$ | Unit holding pollution emission in recycling center $m$ |
| $EH_n$ | Unit holding pollution emission in disposal center $n$ |
| $FJ_i$ | Fixed job opportunity for manufacturing center $i$ |
| $FJ_j$ | Fixed job opportunity for distribution center $j$ |

| | |
|---|---|
| $FJ_m$ | Fixed job opportunity for recycling center $m$ |
| $FJ_n$ | Fixed job opportunity for disposal center $n$ |
| $VJ_i$ | Variable job opportunity for manufacturing center $i$ |
| $VJ_j$ | Variable job opportunity for distribution center $j$ |
| $VJ_m$ | Variable job opportunity for recycling center $m$ |
| $VJ_n$ | Variable job opportunity for disposal center $n$ |
| $FL_i$ | Fixed lost working days because of work damages in manufacturing center $i$ |
| $FL_j$ | Fixed lost working days because of work damages in distribution center $j$ |
| $FL_m$ | Fixed lost working days because of work damages in recycling center $m$ |
| $FL_n$ | Fixed lost working days because of work damages in disposal center $n$ |
| $VL_i$ | Variable lost working days because of work damages in manufacturing center $i$ |
| $VL_j$ | Variable lost working days because of work damages in distribution center $j$ |
| $VL_m$ | Variable lost working days because of work damages in recycling center $m$ |
| $VL_n$ | Variable lost working days because of work damages in disposal center $n$ |
| $PCap_i$ | Processing capacity of manufacturing center $i$ |
| $PCap_m$ | Processing capacity of recycling center $m$ |
| $PCap_n$ | Processing capacity of disposal center $n$ |
| $HCap_j$ | Holding capacity of distribution center $j$ |
| $HCap_m$ | Holding capacity of recycling center $m$ |
| $TMC_v$ | Capacity of transportation mode $v$ |
| $De_k$ | Product demand of customer $k$ |
| $w$ | The weight of product |

Decision variables

| | |
|---|---|
| $X_{ijv}$ | Number of products transited from manufacturing center $i$ to distribution center $j$ by transportation mode $v$ |
| $X_{jkv}$ | Number of products transited from distribution center $j$ to customer $k$ by transportation mode $v$ |
| $X_{kmv}$ | Number of products transited from customer $k$ to recycling center $m$ by transportation mode $v$ |
| $X_{mkv}$ | Number of products transited from recycling center $m$ to customer $k$ by transportation mode $v$ |
| $X_{mnv}$ | Number of products transited from recycling center $m$ to disposal center $n$ by transportation mode $v$ |

$$Y_i \begin{cases} 1 & \text{if manufacturing center } i \text{ is opened} \\ 0 & \text{otherwise} \end{cases}$$

$$Y_j \begin{cases} 1 & \text{if distribution center } j \text{ is opened} \\ 0 & \text{otherwise} \end{cases}$$

$$Y_m \begin{cases} 1 & \text{if recycling center } m \text{ is opened} \\ 0 & \text{otherwise} \end{cases}$$

$$Y_n \begin{cases} 1 & \text{if disposal center } n \text{ is opened} \\ 0 & \text{otherwise} \end{cases}$$

$$Z_{ijv} \begin{cases} 1 & \text{if transportation mode } v \text{ is selected from manufacturing center } i \\ & \text{and distribution center } j \\ 0 & \text{otherwise} \end{cases}$$

$$Z_{jkv} \begin{cases} 1 & \text{if transportation mode } v \text{ is selected from distribution center } j \\ & \text{to customer } k \\ 0 & \text{otherwise} \end{cases}$$

$$Z_{kmv} \begin{cases} 1 & \text{if transportation mode } v \text{ is selected from customer } k \\ & \text{to recycling center } m \\ 0 & \text{otherwise} \end{cases}$$

$$Z_{mkv} \begin{cases} 1 & \text{if transportation mode } v \text{ is selected from recycling center } m \\ & \text{to customer } k \\ 0 & \text{otherwise} \end{cases}$$

$$Z_{mnv} \begin{cases} 1 & \text{if transportation mode } v \text{ is selected from recycling center } m \\ & \text{to disposal center } n \\ 0 & \text{otherwise} \end{cases}$$

### 3.3. Model Formulation

#### 3.3.1. Economic Objective Function

The first optimization objective indicates the economic performance of the SSCN, which aims to maximize total profit. The total profit consists of revenue, inventory holding cost, transportation cost, processing cost and recycling cost. All components of total profit represent economic indicators of the SSCN. The calculation of total profit is calculated by Equation (1).

$$Max\ Z1 = Crev - (Ctrans + Cinv + Cpro + Crec) \tag{1}$$

$$Crev = \sum_{i,j,v} PR_i X_{ijv} + \sum_{m,k,v} PR_m X_{mkv} \tag{2}$$

$$Ctrans = \sum_{i,j,v} CT_{ijv} X_{ijv} + \sum_{j,k,v} CT_{jkv} X_{jkv} + \sum_{k,m,v} CT_{kmv} X_{kmv} \\ + \sum_{m,k,v} CT_{mkv} X_{mkv} + \sum_{m,n,v} CT_{mnv} X_{mnv} \tag{3}$$

$$Cinv = \sum_{i,j,v} CH_i X_{ijv} + \sum_{j,k,v} CH_j X_{jkv} + \sum_{k,m,v} CH_m X_{kmv} + \sum_{m,n,v} CH_n X_{mnv} \tag{4}$$

$$Cpro = \sum_{i,j,v} CP_i X_{ijv} + \sum_{j,k,v} CP_j X_{jkv} + \sum_{m,k,v} CP_m X_{mkv} + \sum_{m,m,v} CP_n X_{mnv} \tag{5}$$

$$Crec = \sum_{k,m,v} CR_k X_{kmv} \tag{6}$$

Equation (2) indicates the total revenue of the SSCN, which comes from new products and renewals of the product. Equation (3) represents transportation costs between two different kinds of facilities. It is should be mentioned that there is forward product flow and reverse product flow between customer markets and recycling centers. Equation (4) indicates the inventory cost involving manufacturing centers, distribution centers, disposal centers and recycling centers. Furthermore, relation Equation (5) calculates the total processing cost involving manufacturing centers, distribution centers and recycling centers. Equation (6) represents the recycling cost from customer markets.

#### 3.3.2. Environmental Objective Function

The second optimization objective represents the environmental performance of the SSCN involving transportation pollution emission, processing pollution emission and holding pollution emission. Each kind of pollution emission represents an environmental indicator.

$$Min\ Z2 = Etrans + Epro + Ehold \tag{7}$$

$$Etrans = \sum_{i,j,v} ET_{ijv} X_{ijv} + \sum_{j,k,v} ET_{jkv} X_{jkv} + \sum_{k,m,v} ET_{kmv} X_{kmv} \\ + \sum_{m,k,v} ET_{mkv} X_{mkv} + \sum_{m,n,v} ET_{mnv} X_{mnv} \tag{8}$$

$$Ehold = \sum_{i,j,v} EH_i X_{ijv} + \sum_{j,k,v} EH_j X_{jkv} + \sum_{k,m,v} EH_m X_{kmv} + \sum_{m,n,v} EH_m X_{mnv} \tag{9}$$

$$Epro = \sum_{i,j,v} EP_i X_{ijv} + \sum_{j,k,v} EP_j X_{jkv} + \sum_{m,k,v} EP_m X_{mkv} + \sum_{m,n,v} EP_n X_{mnv} \tag{10}$$

Equation (8) calculates the total transportation pollution emission. Equation (9) represents the total inventory holding pollution emission of corresponding facilities involving manufacturing centers, distribution centers and recycling centers. Equation (10) indicates the total processing pollution emission that is related to manufacturing centers, distribution centers, disposal centers and recycling centers.

### 3.3.3. Social Objective Function

Social responsibility is an important dimension of the triple bottom line, whose quantitative indicators are difficult to determine. From the perspective of stakeholders and contracts, social responsibility is defined as the fact that organizations or enterprises should take responsibility for the environment, stakeholders and so on in daily operation and business activities [38,39]. The main and most important aspect of the organization's social responsibility is the interests of employees.

In this article, the evaluation indicators of social responsibility are determined based on the established standard ISO2600. Pishvaee et al. measured enterprise social responsibility based on the interests of different stakeholders according to ISO2600 [40]. In ISO2600, the two indicators related to the interests of employees include: (1) lost working days caused by work damage and (2) the number of created job opportunities. The first one reflects the damage caused to employees by working conditions, and the second indicator reflects the positive impact on working conditions and on community development.

Hence, the third objective function evaluates social performance, which aims to maximize job opportunities and minimize lost working days caused by work damages. Both of the aforementioned social factors are considered social evaluation indicators. However, two indicators are in different directions, so $\xi$ and $\zeta$ are adopted as the components of the normalization of weight coefficients to job opportunities and missing work days, respectively.

$$Max \; Z3 = \xi \times JobO - \zeta \times LosD \tag{11}$$

$$
\begin{aligned}
JobO = \;& \sum_i FJ_iY_i + \sum_j FJ_jY_j + \sum_m FJ_mY_m + \sum_n FJ_nY_n \\
& + \sum_{i,j,v} VJ_iX_{ijv}/Pcap_i + \sum_{i,j,v} VJ_jX_{ijv}/Hcap_j + \sum_{m,k,v} VJ_mX_{mkv}/Pcap_i \\
& + \sum_{k,m,v} VJ_mX_{kmv}/Hcap_m + \sum_{m,n,v} VJ_nX_{mnv}/Pcap_n
\end{aligned}
\tag{12}
$$

$$
\begin{aligned}
LosD = \;& \sum_i FL_iY_i + \sum_j FL_jY_j + \sum_m FL_mY_m + \sum_n FL_nY_n \\
& + \sum_i VL_iX_{ijv}/Pcap_i + \sum_j VL_jX_{ijv}/Hcap_j + \sum_m VL_mX_{kmv}/Hcap_m \\
& + \sum_m VL_mX_{mkv}/Pcap_m + \sum_n VL_nX_{mnv}/Pcap_n
\end{aligned}
\tag{13}
$$

The total job opportunities are calculated by Equation (12). The first four terms represent the total fixed job opportunities due to opening the corresponding network facilities, while the surplus terms denote the total created variable job opportunities on the basis of capacity constraints in the corresponding facilities.

The total lost working days are formulated as Equation (13), where the first four parts represent the number of lost working days due to the damages in opening the corresponding network facilities. The remaining parts are related to lost working days resulting from the utilization of facilities' capacities.

### 3.3.4. Constraints

Equations (14)–(17) are flow balance constraints. Equation (14) indicates that the product inflow of distribution centers should be equal to the outflow. Equations (15) and (16) denote the product inflow of recycling centers divided into the product to be remanufactured and disposed



of. Equation (17) represents the number of recycled products that should be equal to or less than the product transported to customer markets.

$$\sum_{i,v} X_{ijv} = \sum_{k,v} X_{jkv} \qquad \forall j \tag{14}$$

$$\sum_{k,v} X_{mkv} = \eta \sum_{k,v} X_{kmv} \qquad \forall m \tag{15}$$

$$\sum_{n,v} X_{mnv} = (1-\eta) \sum_{k,v} X_{kmv} \qquad \forall m \tag{16}$$

$$\sum_{m,v} X_{kmv} \le \sum_{j,v} X_{jkv} \qquad \forall k \tag{17}$$

Equation (18) denotes that the product inflow of customer markets should satisfy the demands.

$$\sum_{j,v} X_{jkv} + \sum_{m,v} X_{mkv} \ge De_k \tag{18}$$

Constraints (19)–(21) are processing capacity constraints involving manufacturing centers, recycling centers and disposal centers.

$$\sum_{j,v} X_{ijv} \le Y_i PCap_i \qquad \forall i \tag{19}$$

$$\sum_{k,v} X_{mkv} \le Y_m PCap_m \qquad \forall m \tag{20}$$

$$\sum_{n,v} X_{mnv} \le Y_n PCap_n \qquad \forall n \tag{21}$$

Equations (22) and (23) are holding capacity constraints that are associated with distribution centers and recycling centers.

$$\sum_{i,v} X_{ijv} \le Y_j HCap_j \qquad \forall j \tag{22}$$

$$\sum_{k,v} X_{kmv} \le Y_m HCap_m \qquad \forall m \tag{23}$$

Constraints (24)–(28) are transportation mode capacity constraints.

$$\sum_{i,j} w X_{ijv} \le \sum_{i,j} Z_{ijv} TMC_v \qquad \forall v \tag{24}$$

$$\sum_{j,k} w X_{jkv} \le \sum_{j,k} Z_{jkv} TMC_v \qquad \forall v \tag{25}$$

$$\sum_{k,m} w X_{kmv} \le \sum_{k,m} Z_{kmv} TMC_v \qquad \forall v \tag{26}$$

$$\sum_{m,k} w X_{mkv} \le \sum_{m,k} Z_{mkv} TMC_v \qquad \forall v \tag{27}$$

$$\sum_{m,n} w X_{mnv} \le \sum_{m,n} Z_{mnv} TMC_v \qquad \forall v \tag{28}$$

Equations (29)–(35) represent that the transportation links between any two facilities are established when the corresponding facilities are opened.

$$Z_{ijv} \le Y_i \qquad \forall i,j,v \tag{29}$$

$$Z_{ijv} \le Y_j \qquad \forall i,j,v \tag{30}$$

$$Z_{jkv} \leq Y_j \qquad \forall j, k, v \tag{31}$$

$$Z_{kmv} \leq Y_m \qquad \forall k, m, v \tag{32}$$

$$Z_{mkv} \leq Y_m \qquad \forall m, k, v \tag{33}$$

$$Z_{mnv} \leq Y_m \qquad \forall m, n, v \tag{34}$$

$$Z_{mnv} \leq Y_n \qquad \forall m, n, v \tag{35}$$

Equations (36) and (37) indicate the characteristics of decision variables.

$$X_{ijv}, X_{jkv}, X_{kmv}, X_{mkv}, X_{mnv} \geq 0 \qquad \forall i, j, k, m, n, v \tag{36}$$

$$Y_i, Y_j, Y_m, Y_n, Z_{ijv}, Z_{jkv}, Z_{kmv}, Z_{mkv}, Z_{mnv} \in \{0, 1\} \qquad \forall i, j, k, m, n, v \tag{37}$$

## 4. Performance-Oriented Optimization Framework

### 4.1. The Basic Framework

To solve the proposed SSCN optimization model, we developed a performance-oriented optimization framework that contains a performance sorting strategy and an optimization solution strategy. In the performance sorting strategy, entropy weight TOPSIS is utilized to measure sustainability performance based on the triple bottom line. In the optimization solution strategy, meta-heuristic algorithms are used to obtain a set of Pareto optimal solutions. The flow chart of the performance-oriented optimization framework is presented in Figure 2, whose essential steps are shown as follows.

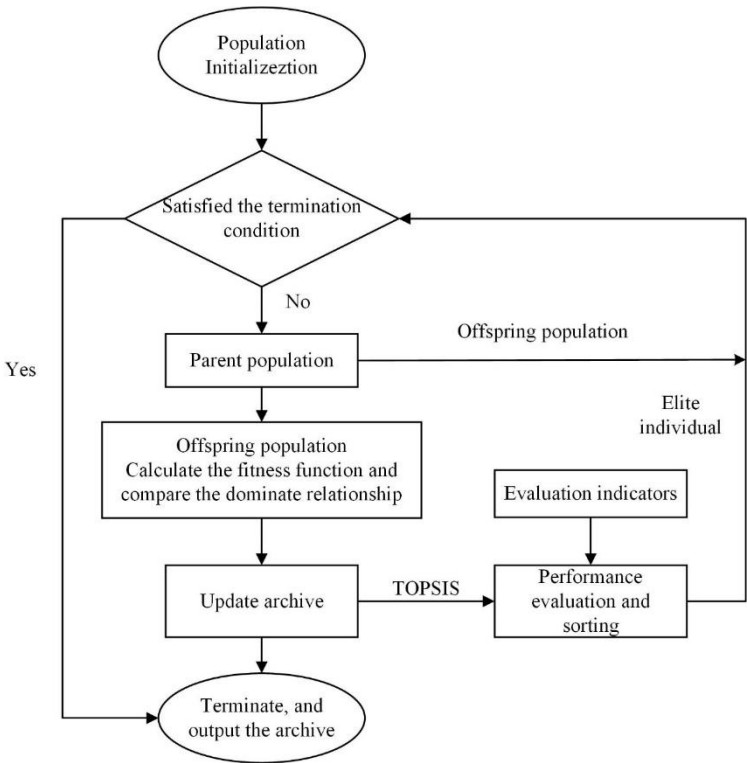

**Figure 2.** Flow chart of the proposed performance-oriented optimization framework.

Step 1: Population initialization
Initialize the population, archive and parameters of algorithms.
Step 2: population update
After initializing the parent population, generate the offspring population based on a special evolution mechanism of different meta-heuristic algorithms. Then, calculate the fitness function values of each individual in the offspring population. Furthermore,

according to the fitness function, compare the non-dominant relationship of individuals and divide them into different layers. The first layer is regarded as the archive. In each algorithm, the elite individual guides the evolution, which is selected according to the supply chain performance evaluation strategy.

Step 3: archive update

Compare the dominant relationship between any two individuals in a mixed population, which is made up of the offspring population and the archive. According to the dominant relationship, the non-dominant individuals are considered the final archive.

Step 4: Performance evaluation and sorting

The entropy weight TOPSIS method is utilized to calculate the performance evaluation value of each individual in the archive. All the Pareto optimal solutions are sorted based on their evaluation value, and the individual with the best performance is selected as the elite individual. Furthermore, the elite individual will be used to guide the population's evolution.

When the algorithm iteration ends, all the individuals in the archive are considered as the Pareto optimal solutions, and the one with the best evaluation value is the optimal scheme.

*4.2. Performance Sorting Strategy*

In Section 3, optimization objectives and evaluation indicators of the sustainable supply chain are developed according to the triple bottom line. The economic indicators consist of revenue, inventory cost, transportation cost, facility processing cost and recycling cost. The environmental indicators include transportation pollution emission, inventory pollution emission and processing pollution emission. The social indicators consist of lost working days caused by work damage and created job opportunities. Among the above-mentioned sustainability indicators, the job opportunities and revenue are as large as possible, while other indicators are as small as possible. On the basis of the established sustainability evaluation indicators system, the entropy weight TOPSIS is utilized to evaluate the sustainability performance of each scheme.

The decision matrix $D$ is formulated, which consists of decision-making units (DMU) and performance evaluation indicators (PEI).

$$D = \left( d_{ij} \right)_{m \times n} = \begin{bmatrix} d_{11} & \cdots & d_{1n} \\ \vdots & \ddots & \vdots \\ d_{m1} & \cdots & d_{mn} \end{bmatrix}, i = 1, 2, \ldots, m; j = 1, 2, \ldots, n \quad (38)$$

where the subscript $n$ represents the number of performance evaluation indicators and $m$ states the number of decision making units. $d_{ij}$ denotes the value of performance evaluation indicator $j$ in decision making unit $i$.

4.2.1. Standardization and Normalization

According to the analyses of objective functions in Section 3.3, all of SC performance includes contradictory indicators. For example, job opportunities should be as big as possible, while lost working days should be as small as possible. Therefore, the corresponding indicators should be standardized before normalization.

If PEI $j$ is a profit indicator, the standardization method is as the following formula.

$$s_{ij} = d_{ij} / \max_i \left( d_{ij} \right), i = 1, 2, \ldots, m; j = 1, 2, \ldots, n \quad (39)$$

If PEI $j$ is a cost indicator, the standardization method is as the listed equation.

$$s_{ij} = \min_i \left( d_{ij} \right) / d_{ij}, i = 1, 2, \ldots, m; j = 1, 2, \ldots, n \quad (40)$$

After all indicators are standardized, the normalization method is as follows.

$$u_{ij} = s_{ij} / \sqrt{\sum_i s_{ij}^2}, i = 1, 2, \ldots, m; j = 1, 2, \ldots, n \quad (41)$$

### 4.2.2. Indicators Weight

The indicator weights in traditional TOPSIS are determined by expert scoring, which is subjective. In order to objectively calculate the indicator weights, the information entropy method is adopted. $e_j$ indicates the entropy value of evaluation indicator $j$ in the standardized decision matrix.

$$e_j = -k \sum_i u_{ij} \ln u_{ij} \tag{42}$$

where $k = 1/\ln m$, $0 \leq e_j \leq 1$. If $b_{ij} = 0$, $u_{ij} \ln u_{ij} = 0$.

$E$ represents the total information entropy of all evaluation indicators.

$$E = \sum_j e_j = -\frac{1}{\ln m} \sum_{i,j} u_{ij} \ln u_{ij} \tag{43}$$

$r_j$ measures the dispersity of evaluation indicator value.

$$r_j = 1 - e_j \tag{44}$$

If the evaluation value of indicator $j$ is more dispersed, the value of $r_j$ is bigger, so the evaluation indicator $j$ is more important. On the contrary, the more concentrated the evaluation value $d_{ij}$ is, the less important the evaluation indicator $j$ is. If all of the evaluation values $d_{ij}$ are equal and absolutely concentrated, the indicator $j$ is invalid. When evaluating the individuals in the archive of a multi-objective optimization algorithm, the objective weight is needed. Therefore, the entropy weights are obtained based on the following methods.

$$w_j = r_j / \sum_j r_j = (1 - e_j) / \sum_j (1 - e_j) = (1 - e_j)/(n - E) \tag{45}$$

The weight factor is calculated by Equation (45).

### 4.2.3. The Weighted Decision Matrix

Based on the standardization decision matrix and the weight vector, the weighted decision matrix can be calculated by the listed relation.

$$V = \begin{bmatrix} w_1 u_{11} & \cdots & w_n u_{1n} \\ \vdots & \ddots & \vdots \\ w_1 u_{m1} & \cdots & w_n u_{mn} \end{bmatrix} = \begin{bmatrix} v_{11} & \cdots & v_{1n} \\ \vdots & \ddots & \vdots \\ v_{m1} & \cdots & v_{mn} \end{bmatrix} \tag{46}$$

### 4.2.4. The Ideal Solution

According to the weighted decision matrix, the positive ideal solution is calculated by Equation (47), while the negative ideal solution is considered as Equation (48).

$$v_j^+ = \begin{cases} \max_i (v_{ij}) & j \in J_1 \\ \min_i (v_{ij}) & j \in J_2 \end{cases} \tag{47}$$

$$v_j^- = \begin{cases} \min_i (v_{ij}) & j \in J_1 \\ \max_i (v_{ij}) & j \in J_2 \end{cases} \tag{48}$$

where $J_1$ denotes the set of profit indicators and $J_2$ is the set of cost indicators.

### 4.2.5. The Relative Distance Calculation

Equation (49) calculates the relative distance between $v_{ij}$ and the positive ideal solution. Equation (50) calculates the relative distance between $v_{ij}$ and the negative ideal solution.

$$Dis_i^+ = \sqrt{\sum_i \left(v_{ij} - v_j^+\right)^2}, i = 1, 2, \ldots, m; j = 1, 2, \ldots, n \tag{49}$$

$$Dis_i^- = \sqrt{\sum_i \left(v_{ij} - v_j^-\right)^2}, i = 1, 2, \ldots, m; j = 1, 2, \ldots, n \tag{50}$$

4.2.6. Performance Evaluation Value

Equation (51) calculates the performance evaluation value.

$$ev_i = Dis_i^- / \left(Dis_i^+ + Dis_i^-\right) \tag{51}$$

According to the performance evaluation value of DMUs calculated by entropy weight TOPSIS, a sequence table can be formulated by sorting the evaluation value.

*4.3. Optimization Solution Strategy*

In this section, a performance-oriented optimization framework is developed. The entropy weight TOPSIS is adopted to measure the supply chain performance of each DMU. Furthermore, the NSGA-II and MOPSO are utilized as examples of formulating the optimization framework.

4.3.1. Fast and Elitist Non-Dominated Sorting Genetic Algorithm (NSGA-II)

NSGA-II is a typical meta-heuristic algorithm based on an individual evolution mechanism. The population evolves through selection, crossover and mutation. Deb et al. introduced an elite non-dominated sorting strategy and elite archive strategy to obtain Pareto optimal solutions [41]. After generating the offspring population based on the above three evolutionary operations, the population are divided into several layers by comparing the dominant relationship. Then, in each layer, all individuals are sorted by calculating crowding distance, whose pseudo-code is presented in Table 1.

**Table 1.** The pseudo-code of crowding distance sorting strategy.

| **Crowding Distance Sorting Strategy** |
| --- |
| Define the initial crowing distance $CD(a_i)$ of all the individuals to zero, <br> for $n = 1 : N_{obj}$, where $N_{obj}$ is the number of fitness functions. <br> Rank the individuals in the Pareto front based on the value of fitness functions, given that $F\max_n$ is the maximum value of fitness functions of all the individuals and $F\min_n$ is the minimum value. <br>   Set the infinite distance to each individual. <br>   $CD(a_1) = \infty$ and $CD(a_{npop}) = \infty$, where $npop$ is the size of the population <br>     for $i = 2 : (npop - 1)$ <br><br>     $CD(a_i) = CD(a_1) + \frac{F_n(x_{i-1}) - F_n(x_{i+1})}{F\max_n - F\min_n}$, where $F_n(x_{i-1})$ is the $n$th t function value of the <br> $(i-1)$th individual, and $F_n(x_{i+1})$ is the $n$th function value of the $(i+1)$th individual. <br><br>     end for <br>   end for |

4.3.2. Multi-Objective Differential Evolution Algorithm (MODE)

MODE is a parallel search algorithm based on biological evolution ideas. The basic and critical steps of the algorithm consist of selection, crossover and mutation. Compared to a genetic algorithm, the mutation mechanism in MODE is based on the differential mutation operation of the parent individuals. The common mutation strategy is shown as follows.

$$y = x_1 + mu \times (x_2 - x_3) \tag{52}$$



where $y$ is the mutation individual and $x_p(p = 1, 2, 3)$ denotes the parent individual. *mu* indicates the mutation factor and $mu \in [0, 1]$. Then, the crossover individual is generated by exchanging some elements of the parent individual and mutation individual as follows.

$$z_i = \begin{cases} y_i & rand_i \leq p_{cross} \\ x_i & otherwise \end{cases} \tag{53}$$

where $i$ indicates the variable in the $i$th dimension of the individual, $p_{cross}$ denotes the crossover possibility and $rand_i$ is a random number that is subject to uniform distribution.

### 4.3.3. Multi-Objective Particle Swarm Optimization (MOPSO) Algorithm

Compared to meta-heuristic algorithms based on the biological evolution mechanism, MOPSO is a typical swarm intelligent optimization algorithm. This algorithm adopts a special particle moving mechanism to converge the particles to the optimal particle. The updating process of location and velocity is the most typical characteristic of this algorithm, which is calculated as follows:

$$v_i(t+1) = wgv_i(t) + c_1 grand_1 g(pbest(t) - x_i(t)) + c_2 grand_2 g(gbest(t) - x_i(t)) \tag{54}$$

$$x_i(t+1) = x_i(t) + v_i(t+1) \tag{55}$$

where $w$ is the inertia weight; $c_1$ and $c_2$ are the cognitive and social coefficients; *pbest* and *gbest* are the personal and global optimal particles; and $t$ indicates the current iteration.

### 4.3.4. Multi-Objective Grey Wolf Optimization (MOGWO) Algorithm

The MOGWO is a typical swarm intelligence algorithm that was developed based on the predation activities of grey wolves. The optimization process of MOGWO consists of social hierarchy, encircling prey, hunting prey, attacking prey and searching for prey. The position of the prey represents the potential optimal position. The grey wolves search for the optimal individual mainly guided by the best three wolves (Alpha, Beta and Delta) in the current population, which is shown in Figure 3. Based on the Pareto-dominated relations, the alpha wolf is superior to the other two wolves, and the beta wolf is better than the delta wolf. According to the position information of the best three wolves, the positions of other search agents (Omega wolf or any other wolves) are updated. The MOGWO searches the optimal solution through the encircling and hunting behavior of grey wolves, which are calculated as follows:

(1)　Encircling

$$\vec{D} = \left| \vec{C} \cdot X_p(t) - \vec{X}(t) \right| \tag{56}$$

$$\vec{X}(t+1) = \vec{X}_p(t) - \vec{A} \cdot \vec{D} \tag{57}$$

where $\vec{X}_p$ indicates the position of the prey and $\vec{X}$ represents the position of the grey wolf. $\vec{A}$ and $\vec{C}$ are the coefficient vectors, which are calculated as follows: $\vec{A} = 2\vec{a} \cdot \vec{r}_1 - \vec{a}$, $\vec{C} = 2 \cdot \vec{r}_2$.

(2)　Hunting

$$\vec{D}_\alpha = \left| \vec{C}_1 \cdot \vec{X}_\alpha - \vec{X} \right|, \vec{D}_\beta = \left| \vec{C}_2 \cdot \vec{X}_\beta - \vec{X} \right|, \vec{D}_\delta = \left| \vec{C}_3 \cdot \vec{X}_\delta - \vec{X} \right| \tag{58}$$

$$\vec{X}_1 = \vec{X}_\alpha - \vec{A}_1 \cdot \vec{D}_\alpha, \vec{X}_2 = \vec{X}_\beta - \vec{A}_2 \cdot \vec{D}_\beta, \vec{X}_3 = \vec{X}_\delta - \vec{A}_3 \cdot \vec{D}_\delta \tag{59}$$

$$\vec{X}(t+1) = \left( \vec{X}_1 + \vec{X}_2 + \vec{X}_3 \right) / 3 \tag{60}$$

According to Equations (58)–(60), when $|A| > 1$, grey wolves are scattered in various areas of search space as far as possible and search for the prey. When $|A| < 1$, the gray wolves concentrate on the prey in one or more areas.

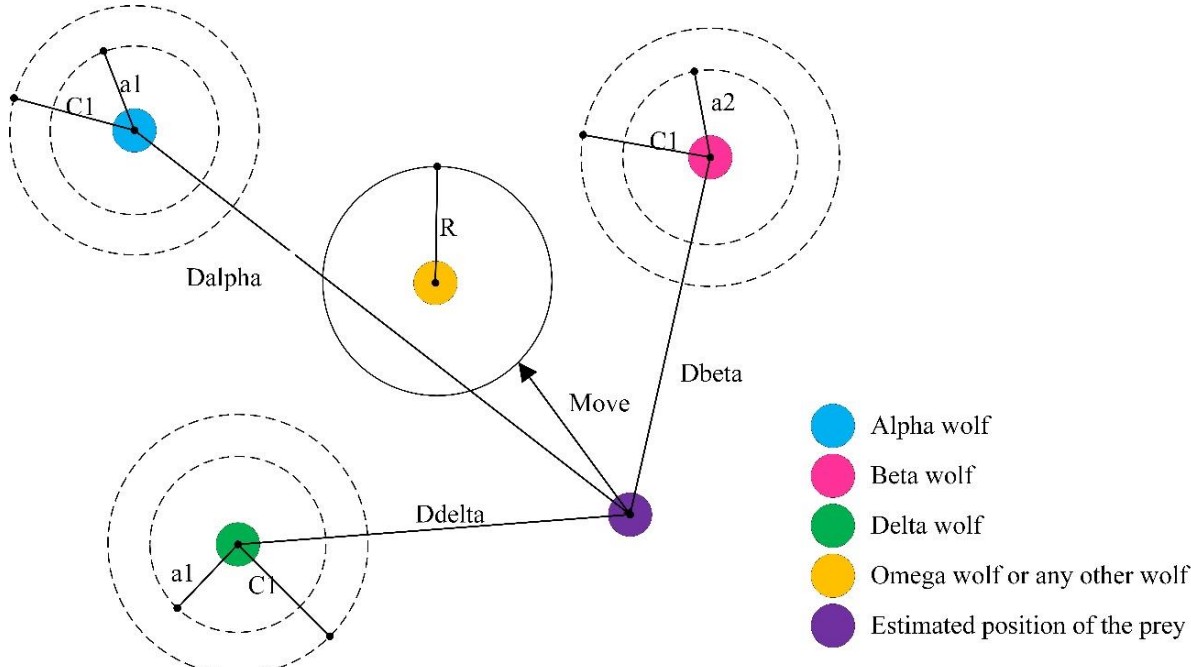

**Figure 3.** Position updating in MOGWO.

## 5. Numerical Case

This section presents a numerical case that is utilized to validate the developed MOMILPM and performance measurement multi-objective optimization framework. The SSCN consists of two manufacturing centers, three distribution centers, four customers, two recycling centers and three distribution centers. The new products are produced in manufacturing centers and transported to distribution centers. Then, they are distributed to each customer from distribution centers. The EOL products are collected and tested by recycling centers. Parts of end-of-life products can be repaired and recovered in recycling centers and returned to the customer market, and the others are delivered to disposal centers. There are two kinds of vehicles used to transport products: general transportation vehicles and new energy transportation vehicles. Compared to general vehicles, the new energy vehicles produce fewer carbon emissions but cost more. The related information on the sustainable supply chain network is shown in Table 2.

**Table 2.** Values of corresponding parameters of the model.

| Parameter | Value | Parameter | Value |
|---|---|---|---|
| $PR_i$ | [50, 60] | $PR_m$ | [45, 48] |
| $CT_{ijv}$, $CT_{jkv}$, $CT_{kmv}$, $CT_{mkv}$, $CT_{mnv}$ | Unif(5, 10) | $CH_i$, $CH_j$, $CH_m$, $CH_n$ | Unif(15, 20) |
| $CP_i$, $CP_j$, $CP_m$, $CP_n$ | Unif(5, 10) | $CR_k$ | Unif(20, 25) |
| $ET_{ijv}$, $ET_{jkv}$, $ET_{kmv}$, $ET_{mkv}$, $ET_{mnv}$ | Unif(5, 10) | $EP_i$, $EP_j$, $EP_m$, $EP_n$ | Unif(5, 10) |
| $EH_i$, $EH_j$, $EH_m$, $EH_n$ | Unif(5, 10) | $FJ_i$, $FJ_j$, $FJ_m$, $FJ_n$ | Unif(150, 200) |
| $VJ_i$, $VJ_j$, $VJ_m$, $VJ_n$ | Unif(150, 200) | $FL_i$, $FL_j$, $FL_m$, $FL_n$ | Unif(15, 25) |
| $VL_i$, $VL_j$, $VL_m$, $VL_n$ | Unif(15, 25) | $PCap_i$, $PCap_m$, $PCap_n$ | Unif(8000, 9000) |
| $HCap_j$, $HCap_m$ | Unif(8000, 9000) | $TMC_v$ | Unif(1000, 2000) |
| $De_k$ | Unif(70, 80) | $w$ | 5 |

### 5.1. Results Analyses

In an attempt to investigate the feasibility and effectiveness of the developed MOMIP model and the performance measurement multi-objective optimization framework, four meta-heuristic algorithms with entropy weight TOPSIS-based evaluation strategy are utilized to solve the mathematical model and are referred to as NSGA-2, MOPSO, MODE, and MOGWO algorithms (EWT-NSGA-2, EWT-MOPSO, EWT-MODE and EWT-MOGWO). In an attempt to control the influence variables of the experiments, the corresponding initial algorithm parameters are presented in Table 3. The numerical case is run 10 times with entropy-weighted TOPSIS-based multi-objective optimization algorithms.

**Table 3.** Initialization parameters of NSGA-2, MODE, MOPSO, and MOGWO.

| Algorithms | NSGA-2 | MODE | MOPSO | MOGWO |
|---|---|---|---|---|
| Population size | 500 | 500 | 500 | 500 |
| Maximum iterations | 1000 | 1000 | 1000 | 1000 |
| Mutation rate | 0.9 | 0.9 | | |
| Crossover rate | 0.8 | 0.8 | | |
| Archive size | | | 200 | 200 |
| Inertia weight | | | 0.7299 | |
| Cognitive and social coefficient | | | 1.4962 | |
| Leader Selection Pressure Parameter | | | | 4 |
| Number of Grids per each Dimension | | | | 10 |
| Grid Inflation Parameter | | | | 0.1 |
| Extra Repository Member Selection Pressure | | | | 2 |

After calculation, by comparing Pareto dominant relations and performance evaluation values, we obtain the final Pareto optimal solutions. Distributions of the Pareto optimal solution set calculated by four meta-heuristic algorithms are shown in Figure 4. In Figure 4, the x-axis represents the economic objective function, the y-axis indicates the environmental objective function and the z-axis states the social objective function. It can be concluded from the distributions that the Pareto optimal solutions of the four algorithms are all uniformly distributed in their search space. The number of solutions obtained by EWT-NSGA-2 and EWT-MOPSO is higher than that of EWT-MODE and EWT-MOGWO. However, because meta-heuristic algorithms solve the multi-objective optimization model based on the random search, the differences in the distribution of Pareto optimal solutions of the four algorithms still exist. On the other hand, the random search mechanism also makes it difficult to distinguish the advantages and disadvantages of these solutions and the dominant relation. Hence, it is essential to further analyze these Pareto optimal solutions quantitatively.

Tables 4–7 represent the evaluation indicators and the performance evaluation value of the final Pareto optimal solutions calculated by entropy-weighted TOPSIS. Based on the triple bottom line of the SCC, the selection of the indicators is subject to the principle of fully reflecting the economic, environmental and social performance. The indicators involving economic performance include transportation cost, inventory holding cost, facilities processing cost, recycling cost and revenue. The environmental evaluation indicators consist of transportation carbon emission, inventory holding pollution emission and processing pollution emission. The social indicators include lost working days caused by work damages and created job opportunities. Among the abovementioned evaluation indicators, the created job opportunities and revenue are the maximum indicators, while the others are minimum indicators. Evaluation values are calculated by entropy-weighted TOPSIS.

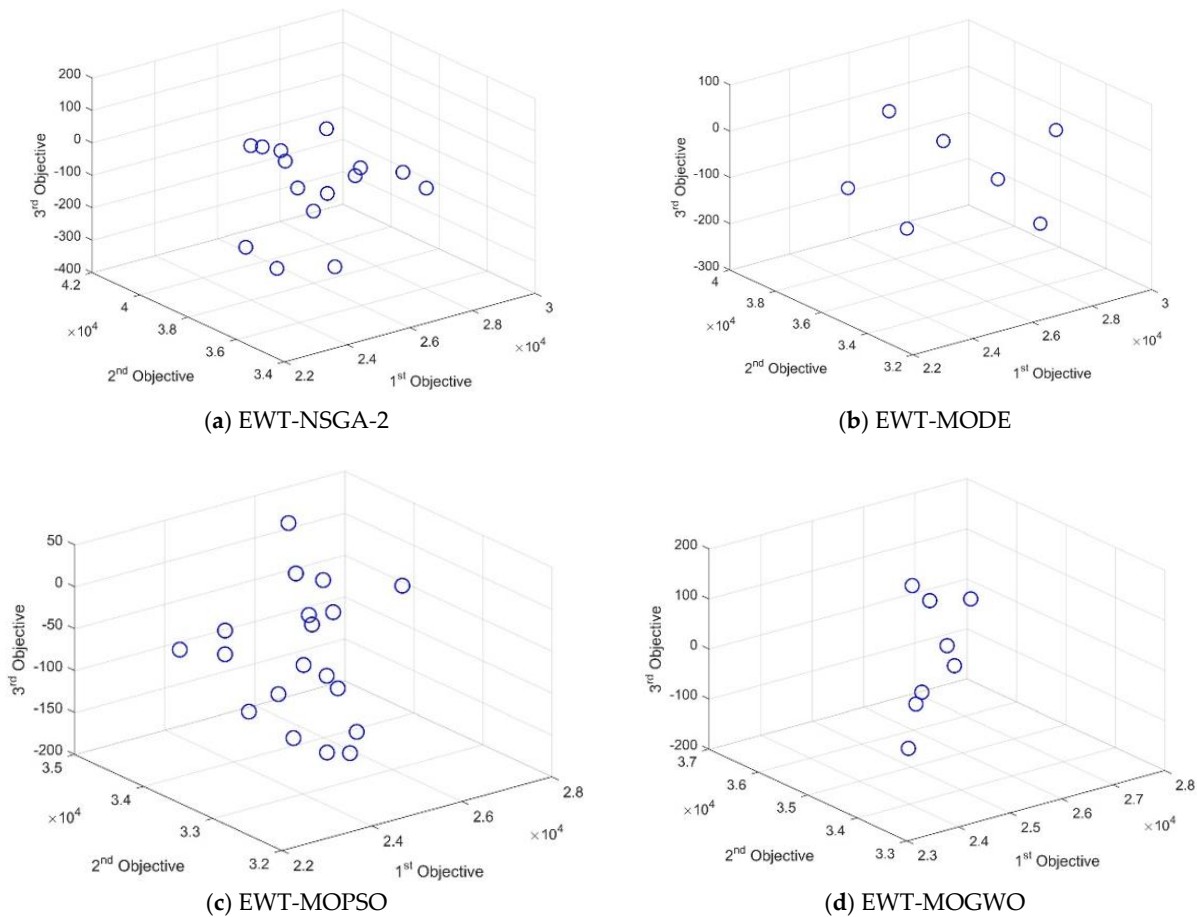

(**a**) EWT-NSGA-2

(**b**) EWT-MODE

(**c**) EWT-MOPSO

(**d**) EWT-MOGWO

**Figure 4.** The distribution of final Pareto optimal solutions obtained by four algorithms.

On the basis of the entropy-weighted method, the weight of each evaluation indicator can be obtained. Because the selection of indicators is guided by the triple bottom line of the SCC, the weight of each dimension is analyzed and compared, which is demonstrated in Figure 5. It can be concluded from the figure that the weight of economic indicators is the largest, which indicates that economic performance plays a leading role in the triple bottom line of the SCC. Furthermore, the social and environmental performance has almost the same influence on the sustainable performance of the supply chain.

**Table 4.** Evaluation indicators and evaluation values of the result of EWT-NSGA-2.

| DMU | Minimum Indicators | | | | | | | | Maximum Indicators | | Evaluation Value |
|---|---|---|---|---|---|---|---|---|---|---|---|
| | Transportation Cost | Inventory Cost | Processing Cost | Recycling Cost | Transportation Emission | Holding Emission | Processing Emission | Lost Working Days | Revenue | Job Opportunities | |
| | Indicators Weight | | | | | | | | | | |
| | 0.11 | 0.06 | 0.13 | 0.09 | 0.06 | 0.16 | 0.07 | 0.13 | 0.05 | 0.14 | |
| 1 | 14,724.73 | 29,898.52 | 13,837.40 | 10,749.65 | 14,344.75 | 13,887.65 | 11,485.77 | 161.20 | 39,677.62 | 1487.51 | 0.8865 |
| 2 | 14,361.68 | 28,563.62 | 13,763.66 | 10,205.04 | 14,589.06 | 12,653.68 | 10,927.86 | 159.74 | 39,677.62 | 1421.82 | 0.5949 |
| 3 | 13,932.28 | 28,527.10 | 14,455.33 | 10,423.81 | 13,797.14 | 11,799.13 | 11,221.76 | 176.82 | 39,677.69 | 1634.16 | 0.5546 |
| 4 | 14,656.03 | 28,003.41 | 13,366.75 | 11,399.21 | 14,310.72 | 12,323.26 | 11,480.53 | 185.17 | 39,677.69 | 1634.27 | 0.5472 |
| 5 | 14,070.74 | 28,816.63 | 13,393.86 | 10,778.25 | 14,435.16 | 12,385.55 | 10,343.62 | 170.97 | 39,677.64 | 1454.93 | 0.5225 |
| 6 | 14,059.53 | 26,451.98 | 14,325.75 | 10,859.70 | 14,150.35 | 11,902.80 | 11,154.19 | 167.00 | 39,677.63 | 1404.28 | 0.5212 |
| 7 | 14,344.86 | 29,566.65 | 12,141.93 | 10,735.51 | 14,205.79 | 12,441.38 | 10,552.93 | 165.59 | 39,677.62 | 1469.18 | 0.4130 |
| 8 | 14,677.20 | 28,801.65 | 12,320.07 | 11,075.90 | 14,489.32 | 11,909.75 | 11,259.94 | 177.86 | 39,677.69 | 1625.41 | 0.3990 |
| 9 | 14,600.93 | 29,315.84 | 12,139.80 | 10,883.61 | 14,246.32 | 12,128.20 | 11,086.38 | 168.38 | 39,677.52 | 1478.81 | 0.3987 |
| 10 | 14,199.48 | 27,639.80 | 12,051.31 | 11,176.67 | 14,542.70 | 12,302.48 | 12,132.20 | 168.00 | 39,677.65 | 1475.56 | 0.3852 |
| 11 | 14,797.54 | 29,774.89 | 11,401.13 | 11,027.97 | 14,102.98 | 12,540.79 | 10,495.07 | 162.38 | 39,677.61 | 1538.99 | 0.3768 |
| 12 | 14,890.58 | 30,405.61 | 10,849.30 | 11,184.85 | 14,095.77 | 12,092.77 | 10,879.46 | 170.19 | 39,677.62 | 1424.98 | 0.3466 |
| 13 | 14,581.27 | 27,850.74 | 11,811.93 | 10,521.09 | 14,521.72 | 12,634.31 | 10,344.06 | 175.28 | 39,677.69 | 1597.48 | 0.3368 |
| 14 | 15,206.74 | 28,856.92 | 10,919.89 | 10,581.38 | 14,406.65 | 12,540.09 | 10,515.66 | 182.81 | 39,677.69 | 1610.47 | 0.3221 |
| 15 | 14,347.65 | 28,837.43 | 12,218.86 | 10,373.78 | 14,366.42 | 11,736.28 | 9696.27 | 177.63 | 39,677.69 | 1585.83 | 0.3173 |

**Table 5.** Evaluation indicators and evaluation values of the result of EWT-MODE.

| DMU | Minimum Indicators | | | | | | | | Maximum Indicators | | Evaluation Value |
|---|---|---|---|---|---|---|---|---|---|---|---|
| | Transportation Cost | Inventory Cost | Processing Cost | Recycling Cost | Transportation Emission | Holding Emission | Processing Emission | Lost Working Days | Revenue | Job Opportunities | |
| | Indicators Weight | | | | | | | | | | |
| | 0.09 | 0.12 | 0.09 | 0.09 | 0.10 | 0.08 | 0.11 | 0.12 | 0.09 | 0.11 | |
| 1 | 13,939.98 | 28,989.29 | 12,762.36 | 11,064.01 | 14,135.09 | 12,837.14 | 10,049.19 | 188.97 | 38,453.00 | 1796.97 | 0.7535 |
| 2 | 14,196.11 | 28,636.41 | 12,955.28 | 11,541.15 | 14,881.02 | 11,089.41 | 9848.86 | 209.27 | 38,394.00 | 1575.75 | 0.5872 |
| 3 | 13,742.01 | 28,144.53 | 11,938.37 | 10,825.11 | 13,708.81 | 11,280.24 | 10,746.18 | 215.47 | 38,208.00 | 1856.64 | 0.5275 |
| 4 | 13,558.88 | 26,889.80 | 11,865.69 | 10,482.14 | 13,740.00 | 13,165.77 | 11,957.38 | 149.89 | 38,039.00 | 1370.14 | 0.3393 |
| 5 | 13,855.49 | 27,587.60 | 12,579.15 | 9943.16 | 12,700.14 | 11,537.24 | 10,189.36 | 153.49 | 37,663.00 | 1427.07 | 0.7070 |
| 6 | 13,795.81 | 26,653.46 | 12,709.03 | 10,586.51 | 13,331.72 | 11,544.01 | 9935.29 | 196.39 | 37,105.00 | 1793.63 | 0.6445 |
| 7 | 13,909.61 | 27,631.68 | 11,426.34 | 10,717.19 | 13,397.26 | 9784.08 | 9191.08 | 207.11 | 37,576.00 | 1869.41 | 0.3504 |

**Table 6.** Evaluation indicators and evaluation values of the result of EWT-MOPSO.

| DMU | Minimum Indicators | | | | | | | | Maximum Indicators | | Evaluative Value |
| | Transportation Cost | Inventory Cost | Processing Cost | Recycling Cost | Transportation Emission | Holding Emission | Processing Emission | Lost Working Days | Revenue | Job Opportunities | |
| | Indicators Weight | | | | | | | | | | |
| | 0.10 | 0.07 | 0.14 | 0.08 | 0.09 | 0.09 | 0.07 | 0.11 | 0.04 | 0.22 | |
| **1** | **13,333.96** | **26,196.12** | **11,495.71** | **10,696.50** | **13,435.46** | **12,055.04** | **10,612.45** | **80.35** | **35,051.00** | **718.51** | **0.8191** |
| 2 | 12,725.70 | 26,101.28 | 12,602.84 | 10,470.25 | 13,195.01 | 12,117.34 | 10,227.88 | 98.17 | 35,412.00 | 932.25 | 0.7267 |
| 3 | 13,328.37 | 25,557.21 | 10,898.89 | 9891.87 | 13,562.62 | 11,574.82 | 9641.73 | 79.05 | 35,159.00 | 725.60 | 0.6358 |
| 4 | 12,759.50 | 26,231.09 | 11,710.94 | 10,909.18 | 12,505.41 | 12,530.63 | 9222.37 | 91.80 | 35,051.00 | 725.03 | 0.6266 |
| 5 | 13,294.91 | 27,316.45 | 10,518.21 | 10,610.48 | 13,676.42 | 11,510.63 | 9245.45 | 81.80 | 35,147.00 | 737.04 | 0.5473 |
| 6 | 12,784.99 | 25,180.15 | 11,262.00 | 10,314.70 | 14,008.49 | 11,286.81 | 9721.50 | 99.43 | 35,355.00 | 935.41 | 0.5462 |
| 7 | 13,041.41 | 25,536.22 | 10,395.48 | 10,299.27 | 13,620.61 | 11,187.26 | 9922.26 | 87.19 | 35,149.00 | 771.63 | 0.5180 |
| 8 | 13,006.47 | 26,397.21 | 11,080.56 | 10,483.44 | 12,740.59 | 10,936.56 | 9254.18 | 104.42 | 35,412.00 | 927.92 | 0.4364 |
| 9 | 12,863.93 | 25,357.85 | 10,302.66 | 10,326.86 | 13,503.35 | 11,460.39 | 10,280.31 | 77.23 | 35,147.00 | 744.44 | 0.4201 |
| 10 | 13,311.10 | 26,801.62 | 9739.46 | 9906.41 | 13,128.22 | 12,065.88 | 9890.65 | 91.27 | 35,159.00 | 741.98 | 0.4133 |
| 11 | 12,810.66 | 26,344.41 | 11,197.97 | 10,391.45 | 12,818.32 | 10,612.50 | 9871.11 | 96.93 | 35,147.00 | 711.86 | 0.4094 |
| 12 | 12,700.18 | 25,264.54 | 11,161.52 | 10,897.25 | 13,711.56 | 9700.43 | 9542.20 | 97.83 | 35,144.00 | 839.29 | 0.3996 |
| 13 | 12,874.01 | 25,388.72 | 11,577.28 | 10,092.38 | 13,106.03 | 10,501.45 | 8803.32 | 85.77 | 35,051.00 | 759.94 | 0.3795 |
| 14 | 12,875.37 | 25,970.89 | 11,012.95 | 10,369.57 | 13,211.26 | 9158.74 | 8565.61 | 100.36 | 35,412.00 | 890.89 | 0.3702 |
| 15 | 13,675.22 | 25,491.12 | 10,011.61 | 10,688.65 | 12,938.79 | 9814.93 | 9396.28 | 85.94 | 35,111.00 | 757.05 | 0.3451 |
| 16 | 12,817.27 | 25,605.42 | 9998.83 | 9638.21 | 12,726.34 | 11,768.65 | 9330.87 | 99.04 | 35,355.00 | 839.15 | 0.3231 |
| 17 | 12,812.90 | 25,358.07 | 10,588.46 | 10,459.20 | 13,029.05 | 9406.98 | 9758.30 | 87.51 | 34,379.00 | 770.12 | 0.3094 |
| 18 | 13,297.80 | 24,567.79 | 9789.91 | 9786.23 | 13,486.21 | 10,612.81 | 10,046.96 | 92.04 | 35,051.00 | 744.89 | 0.3066 |
| 19 | 12,493.09 | 24,130.95 | 10,242.25 | 10,527.24 | 13,248.25 | 10,766.47 | 9342.81 | 101.51 | 35,415.00 | 918.55 | 0.2111 |

**Table 7.** Evaluation indicators and evaluation values of the result of EWT-MOGWO.

| DMU | Minimum Indicators | | | | | | | | Maximum Indicators | | Evaluative Value |
|---|---|---|---|---|---|---|---|---|---|---|---|
| | Transportation Cost | Inventory Cost | Processing Cost | Recycling Cost | Transportation Emission | Holding Emission | Processing Emission | Lost Working Days | Revenue | Job Opportunities | |
| | Indicators Weight | | | | | | | | | | |
| | 0.08 | 0.17 | 0.08 | 0.11 | 0.06 | 0.10 | 0.11 | 0.14 | 0.08 | 0.09 | |
| 1 | 13,240.40 | 27,658.40 | 11,877.34 | 11,304.27 | 13,939.78 | 10,909.30 | 9866.20 | 123.93 | 36,989.00 | 1063.04 | 0.3607 |
| 2 | 14,176.28 | 27,623.41 | 11,814.75 | 10,510.77 | 12,708.78 | 11,334.77 | 10,099.29 | 109.39 | 37,065.00 | 929.66 | 0.6214 |
| 3 | 13,983.93 | 26,314.16 | 11,189.53 | 10,603.97 | 13,689.62 | 11,305.18 | 9139.43 | 111.26 | 37,012.00 | 912.26 | 0.3470 |
| 4 | 13,678.85 | 27,037.43 | 10,628.18 | 10,040.56 | 13,765.30 | 11,245.05 | 9934.84 | 111.35 | 36,989.00 | 1093.58 | 0.3972 |
| 5 | 13,735.49 | 26,352.19 | 11,221.07 | 10,938.28 | 14,381.12 | 11,773.15 | 10,467.18 | 129.31 | 36,989.00 | 1062.69 | 0.6127 |
| 6 | 13,488.60 | 26,380.91 | 11,436.97 | 9897.41 | 13,624.51 | 12,072.35 | 9201.91 | 131.94 | 36,909.00 | 1099.82 | 0.2443 |
| **7** | **13,760.54** | **26,772.81** | **10,983.38** | **10,975.26** | **13,666.55** | **12,220.23** | **10,883.07** | **120.35** | **36,941.00** | **1104.67** | **0.6896** |
| 8 | 14,264.81 | 26,985.37 | 11,174.11 | 10,248.38 | 13,940.30 | 10,674.86 | 10,872.93 | 132.92 | 36,989.00 | 1141.80 | 0.3152 |

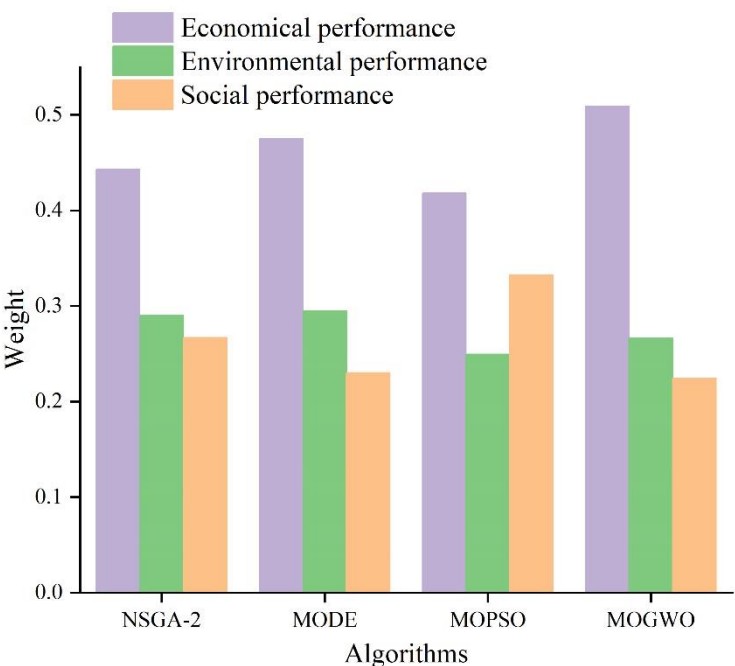

**Figure 5.** Comparison of weights of three sustainability dimensions.

By comparing the performance evaluation value of the Pareto optimal solutions, all the feasible solutions are sorted, and the solution with the best performance evaluation value is considered as the optimal solution, which is bolded in each table. From Tables 4–7 and Figure 6d the performance evaluation value of the optimal solution is 0.8865 in EWT-NSGA-2, 0.7535 in EWT-MODE, 0.8191 in EWT-MOPSO and 0.6896 in EWT-MOGWO. Hence, the solution calculated by EWT-MOPSO and EWT-NSGA-2 has better sustainability performance than the other two meta-heuristic algorithms based on EW-TOPSIS.

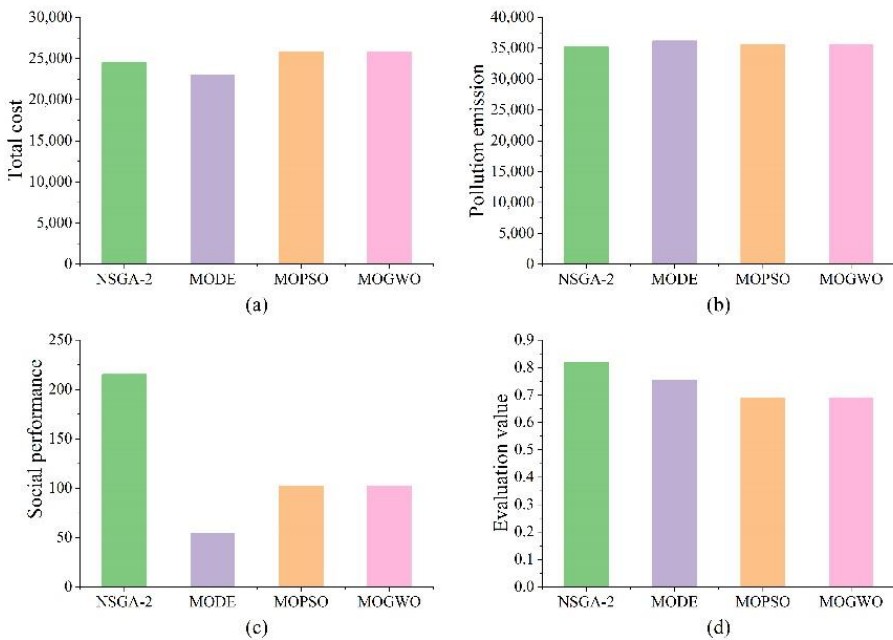

**Figure 6.** Comparison of objective functions and evaluation values obtained by four algorithms.

Figure 6a–c illustrate the comparison of three function values of the optimal solution in the case of four algorithms. Parts (a) and (b) indicate the comparison of total cost and total pollution emission of the sustainable supply chain. It can be concluded that the total

cost and pollution emission of the optimal solution obtained by the four algorithms are almost the same. However, part (c) in Figure 6 indicates that the social performance of the optimal solution obtained by EWT-NSGA-2 is much better than that obtained by the other three algorithms.

### *5.2. Sensitivity Analyses*

5.2.1. Effect of Performance Sorting Strategy on Meta-Heuristic Algorithms

This section performs a sensitivity analysis of different strategies for the performance of the performance-oriented optimization framework. The meta-heuristic algorithms with entropy-weighted TOPSIS, with TOPSIS, and without any evaluation strategy are utilized to calculate the numerical case. Then, we compare the performance evaluation value of the obtained Pareto optimal solution set. In the case of the meta-heuristic algorithms without an evaluation strategy, the final Pareto optimal solutions are evaluated by entropy-weighted TOPSIS so as to measure the sustainability performance.

Figure 7 presents the comparison of the performance evaluation values of four algorithms with the abovementioned three evaluation strategies. In this figure, "EW-TOPSIS" indicates the algorithms with entropy-weighted TOPSIS, "TOPSIS" represents the algorithms with TOPSIS, and "NON" states the meta-heuristic algorithms without evaluation strategy.

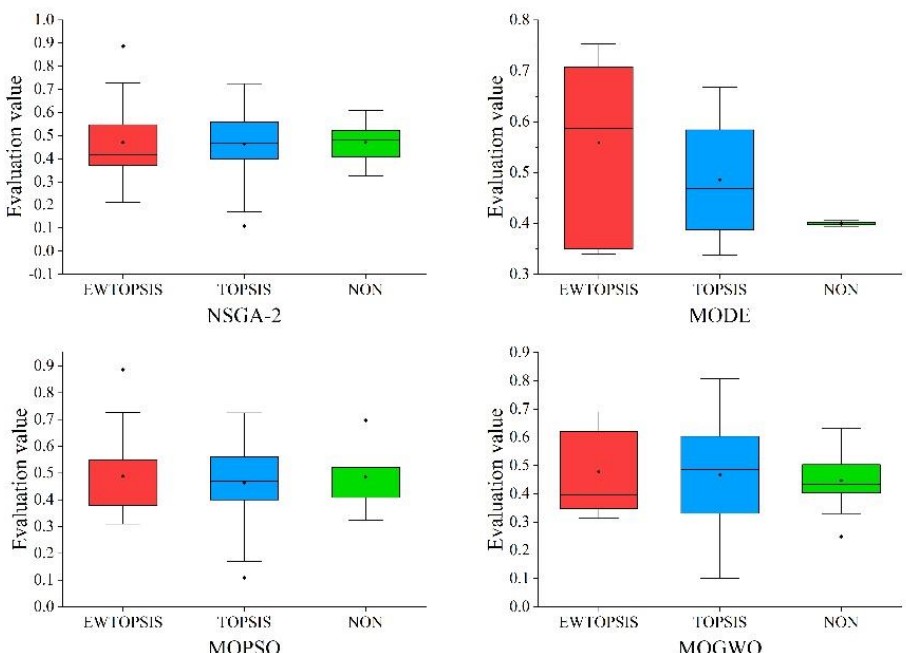

**Figure 7.** Comparison of performance evaluation values based on three evaluation strategies.

Comparing maximum loci and loci greater than 75% of data shows that the performance evaluation values of algorithms with EW-TOPSIS and TOPSIS are generally greater than those of algorithms without an evaluation strategy. On the other hand, comparing the average number of each part in Figure 8 shows that the overall performance of the feasible solutions obtained by algorithms with EW-TOPSIS is better than that obtained by algorithms with TMOOAs and without an evaluation strategy. The above phenomenon occurs because the entropy-weighted TOPSIS method and TOPSIS method can make the population of algorithms converge to the optimal solution with the best performance evaluation value. Furthermore, entropy-weighted TOPSIS weights each indicator objectively according to performance evaluation values and reduces the deviation caused by subjective assignment.

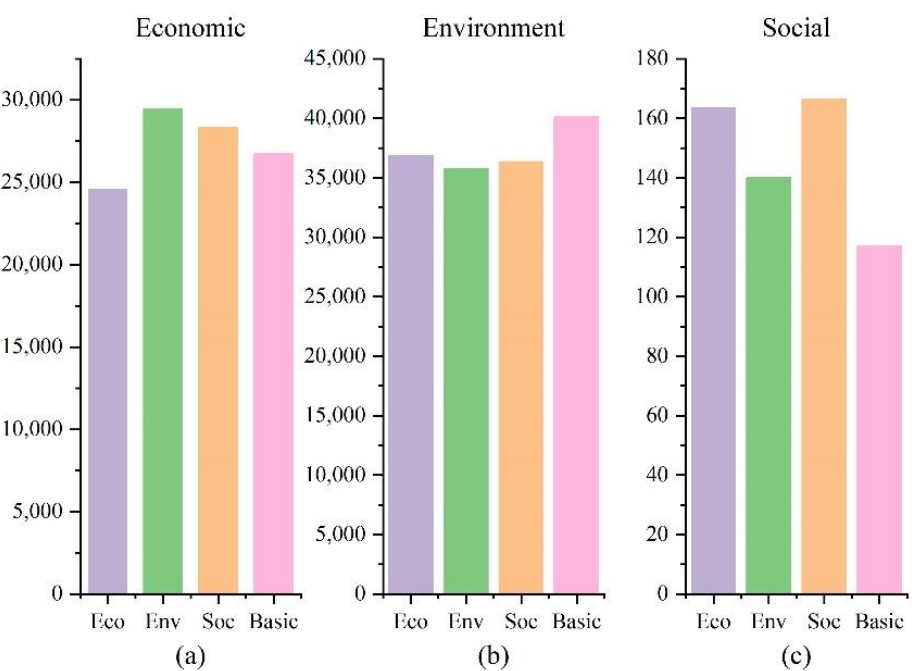

**Figure 8.** Effect of sustainability dimension on objective functions based on EWT-NSGA-2.

5.2.2. Effect of Indicator Dimension on Sustainability Performance

In this section, in an attempt to examine the influence of different dimensions on the optimization objectives, the proposed model is calculated by algorithms with EW-TOPSIS with economic, environmental and social evaluation indicators, respectively. According to the analyses of results in Section 5.1, the solution sets obtained by EWT-NSGA-2 and EWT-MOPSO have better sustainability performance than that obtained by EWT-MODE and EWT-MOGWO. Hence, EWT-NSGA-2 and EWT-MOPSO are utilized to conduct sensitivity analyses. In an attempt to separately analyze the effect of a single sustainability indicator on the objective functions, only one dimension indicator is used to evaluate the sustainability performance of the obtained solution sets. For example, when the social dimension is taken as the research object, only lost working days and created job opportunities are selected as evaluation indicators.

The results of three single sustainability dimensions obtained by EWT-NSGA-2 and EWT-MOPSO are demonstrated in Figures 8 and 9 respectively. The optimal solution with the greatest performance evaluation value is utilized to calculate objective functions. In the two figures, "Eco" indicates that only economic indicators are selected, "Env" represents only environmental indicators, "Soc" states only social indicators and "Basic" represents that all sustainability indicators are used. Part (a) presents the economical objective function, part (b) states the environmental objective function and part (c) shows the social objective function.

It can be seen from part (a) of Figures 8 and 9 that the total cost is the smallest when only economic indicators are considered. Secondly, according to part (b) of Figures 8 and 9, the sustainability supply chain has the smallest pollution emission when only environmental indicators are considered. Thirdly, in part (c) of Figures 8 and 9, it can be seen that when only social indicators are utilized for evaluation, the supply chain has the best social performance. Furthermore, compared to "Basic", the objective function value of this dimension under a single dimension is better than that of the corresponding dimension under all sustainability dimensions. The above phenomenon occurs because when only a single sustainability dimension is used to evaluate Pareto optimal solutions, the meta-heuristic algorithm evolves towards the best performance of this dimension.

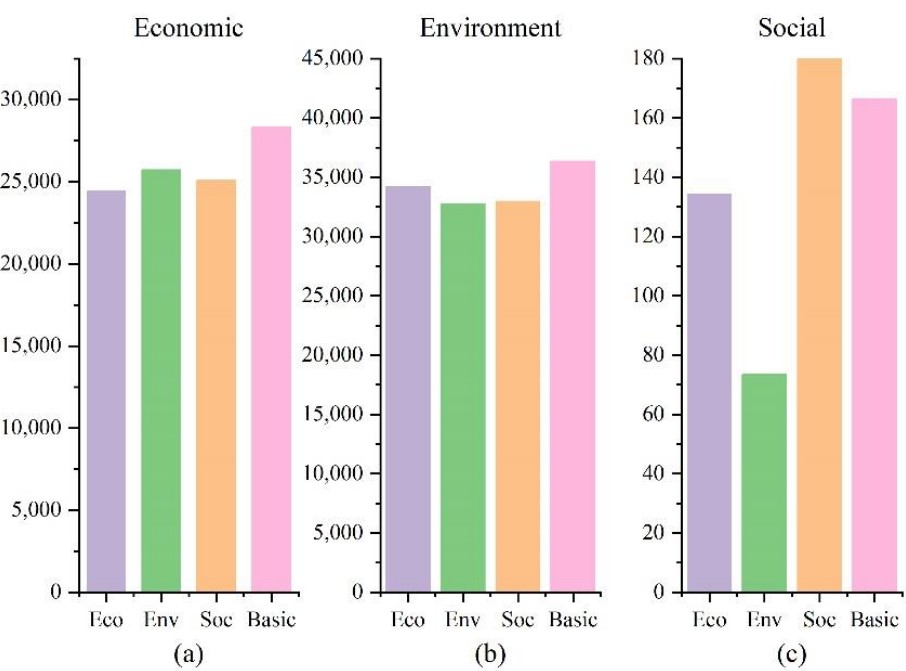

**Figure 9.** Effect of sustainability dimension on objective functions based on EWT-MOPSO.

After the optimal solution by meta-heuristic algorithms with four indicators' selection methods is obtained, the sustainability performance of four optimal solutions is evaluated by the entropy-weighted TOPSIS method. Figure 10 shows the comparison of sustainability performance evaluation values of the abovementioned four indicators' selection methods. It can be concluded from Figure 10 that the sustainability performance of the optimal solution obtained by meta-heuristic algorithms with overall sustainability indicators is much better than that obtained by meta-heuristic algorithms with any single-dimension indicators.

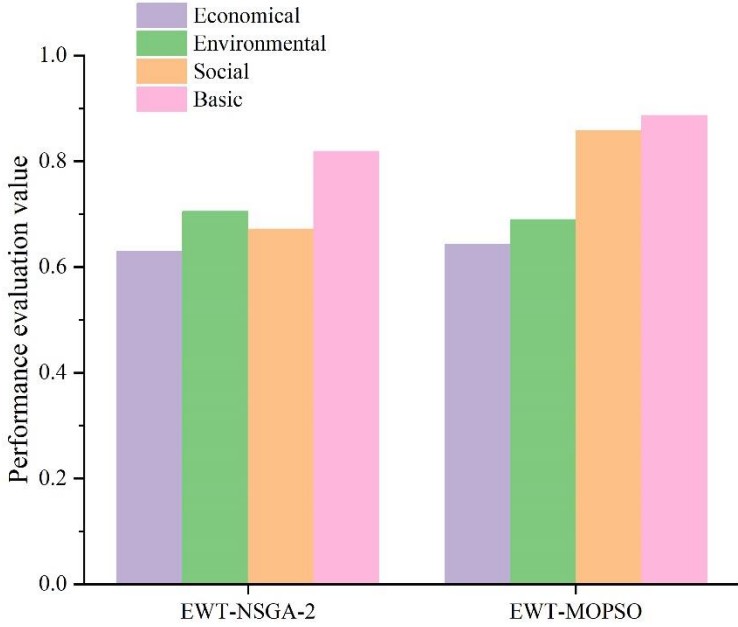

**Figure 10.** Comparison of performance evaluation values of four indicators selection methods.

### 5.3. Managerial Insights

In this section, some practical managerial insights based on the analyses of results and sensitivity analyses are as follows:

- This article proposed a MOMIP model to address the problem of SSCN optimization and design, which provides managers with a good perspective to manage the supply chain and improve sustainability performance. The proposed model can assist managers in decision-making on the transportation mode selection, facilities location, suppliers selection and the flow of products.
- The triple bottom line of the SCC is utilized to determine optimization objectives and select the appropriate evaluation indicators in the proposed model and the solution method. The proposed model and method help supply chain managers achieve better sustainable performance. The results analyses prove that considering sustainability based on the triple bottom line can achieve greater sustainable performance than only considering a single dimension.
- The proposed performance-oriented optimization framework can provide managers with a special optimal scheme with the best sustainable performance. On the one hand, meta-heuristic algorithms are proposed to deal with the optimization problem in this framework. Because SSCN optimization is a complex optimization problem, meta-heuristic algorithms are effective and realizable in solving this kind of problem. On the other hand, entropy-weighted TOPSIS weights each sustainable indicator and sorts the feasible schemes. The entropy weight method can effectively avoid the influence of artificial factors and calculate the objective weights. Furthermore, the selected evaluation indicators should be quantitative and can comprehensively embody the sustainability performance of the supply chain.

## 6. Conclusions

This article addresses an SSCN design problem integrating network optimization and performance evaluation. The optimization objectives and evaluation indicators are determined on the basis of the triple bottom line. From the three dimensions of economy, environment and society, a MOMIP model was developed that aims to minimize the total cost of the SSCN, minimize the environmental pollution and maximize social responsibility (i.e., lost working days caused by work damages and job opportunities). Furthermore, in an attempt to solve the developed mathematical model, this article proposed a performance-oriented optimization framework. This framework integrates supply chain network optimization and sustainable performance evaluation together. This framework adopts meta-heuristic algorithms as the optimization method and entropy-weighted TOPSIS as the evaluation method. Based on this framework, managers can not only obtain feasible schemes but also evaluate the performance of these schemes so as to obtain the optimal scheme. Then, to validate the developed optimization model and propose an optimization framework, four multi-objective meta-heuristic algorithms and the performance evaluation strategy are combined in pairs, and a numerical case is tested.

Moreover, this article performs some sensitivity analyses on the performance evaluation strategy and the sustainability indicators dimension. The results indicate that the optimal solution obtained by algorithms with EW-TOPSIS has better sustainable performance than that obtained by meta-heuristic algorithms with no evaluation strategy or traditional TOPSIS. Finally, some practical managerial insights are proposed based on the abovementioned analyses of results and comparative analyses.

The main contributions of this article are listed as follows:

- Supply chain managers can obtain the feasible Pareto optimal solution set of the optimization model with the performance-oriented optimization framework. The sustainability performance of these solutions can be evaluated and sorted so as to determine the optimal solution in this framework.
- This framework can use the entropy weight method to obtain the objective weight of each performance evaluation indicator.
- The performance evaluation method can guide the evolution process of algorithms so as to make the population migrate to the individual with the greatest sustainability performance.

- This framework can flexibly combine a variety of optimization algorithms and multi-criteria decision-making methods.

  This article also puts forward some research directions to expand the presented problems and theories. The operation of a sustainable supply chain faces many uncertainties and the possibility of interruption due to natural disasters or human-made accidents. Hence, the resilience of the supply chain can be taken into consideration in supply chain network design. Additionally, because of the high complexity and large scale of the model in real-world supply chain management, developing other solution algorithms such as hyper-heuristic algorithms could be a promising avenue for further study. Likewise, additional sustainability evaluation indicators can be considered, and the application of other appropriate qualitative evaluation methods can provide valuable advice for future research.

**Author Contributions:** Conceptualization, Y.G. and Q.S.; methodology, Y.G.; software, Y.G.; validation, Q.S.; formal analysis, Y.G.; investigation, Y.G.; resources, C.G.; data curation, C.G.; writing—original draft preparation, Y.G.; writing—review and editing, C.G.; visualization, Y.G.; supervision, Q.S.; project administration, Q.S.; funding acquisition, C.G. All authors have read and agreed to the published version of the manuscript.

**Funding:** This research was funded by The National Natural Science Foundation of China, grant number 71871220.

**Data Availability Statement:** The data presented in this study are available on request from the corresponding author.

**Acknowledgments:** Our deepest gratitude goes to the anonymous referees for detailed reviews and insightful comments that have helped improve this research substantially. This study was supported by The National Natural Science Foundation of China under grant number 71871220. This support is gratefully appreciated.

**Conflicts of Interest:** The authors declare no conflict of interest.

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
