# Peer review of "A Performance-Oriented Optimization Framework Combining Meta-Heuristics and Entropy-Weighted TOPSIS for Multi-Objective Sustainable Supply Chain Network Design"

_electronics, doi:10.3390/electronics11193134_

Round 1

Reviewer 1 Report

Dear Authors

I appreciate the work you have done. However, I make some suggestions, taking an opportunity for improvement:

1. In the abstract, the authors presented their results, but I believe they could point out the practical implications of their model and indicate opportunities for future research. Such as the use of other multicriteria methods.

2. In lines 152-153, the authors present the following information: "In this article, entropy weight TOPSIS method is utilized to evaluate the sustainability performance of decision schemes". I suggest to the readers to present the reason for the choice of this method. In this sense, I suggest the inclusion of the following paper, which I believe has relevant contributions to the present question: "A Systematic Review of the Applications of Multi-Criteria Decision Aid Methods (1977-2022)". Electronics 2022, 11, 1720. https://doi.org/10.3390/electronics11111720.

3. On line 210, the authors have changed the style of citations in the text. I suggest keeping the style proposed by the journal.

4. In line 395, figure 3 should be considered as a table or chart.

5. Finally, I suggest updating the references with the most recent publications on the subject.

Finally, I congratulate the authors on the work proposed. Good review.

Reviewer.

Reviewer 2 Report

The paper is really good organized.

All the sections are presented extended. Maybe sometimes the content might be better concentrated. I really liked to read this paper. 

The mathematical background of the research is OK.

In section 4

" ......... are shown as follows" - a diagram with those steps could be easier for reading the article.

Figure 2 -  has to be better explained

Section 4.3

Figure 3 - is quite hard to follow from the figure the context

Round 2

Reviewer 1 Report

Dear Authors

Once again, I congratulate you on your excellent work submitted for evaluation. The work is consistent, and the reviewers' suggestions in the first review round have been implemented. In this way, I believe that the existing gaps have been filled. In the current version, I do not see any improvement that can be added. I wish you success in the next round of research.

Best Regards

Reviewer